# Feasibility of Intratumoral Anti-PD1 as Treatment of Human Basal Cell Carcinoma: An Explorative Study with Adjuvant Ablative Fractional Laser

**DOI:** 10.3390/cancers14235815

**Published:** 2022-11-25

**Authors:** Silje Haukali Omland, Jacob Secher Ejlertsen, Dorrit Krustrup, Rikke Louise Christensen, Inge Marie Svane, Uffe Hoegh Olesen, Merete Hædersdal

**Affiliations:** 1Department of Dermatology, Copenhagen University Hospital, Bispebjerg and Frederiksberg, Copenhagen, Denmark; 2Department of Pathology, Copenhagen University Hospital, Herlev and Gentofte, Copenhagen, Denmark; 3Department of Oncology, Center for Cancer Immune Therapy, Copenhagen University Hospital, Herlev and Gentofte, Copenhagen, Denmark; 4Department of Oncology, Copenhagen University Hospital, Herlev, Denmark

**Keywords:** basal cell carcinoma (BCC), keratinocyte carcinoma (KC), immune checkpoint inhibitor (ICI), anti-programmed death-1 (anti-PD1), immune response, ablative fractional CO2 laser (AFL)

## Abstract

**Simple Summary:**

The use of immune checkpoint inhibitors (ICI) is expanding with the approval for locally advanced and metastasizing keratinocyte carcinoma. Most cases, however, are non-aggressive. Systemic therapy remains limited by severe side effects. Local administration could broaden ICI, but an adequate immune response might require an immune-attractive adjuvant such as ablative fractional laser (AFL). Accordingly, aiming to broaden the field of ICI to keratinocyte carcinoma, this study investigates the intratumoral injection of anti-PD1 with and without AFL in basal cell carcinoma (BCC), exploring anti-PD1 concentration, immune cell infiltration, tumor response, and safety. With the results of the study showing the feasibility of intratumoral anti-PD1 and increase in immune cell infiltration and tumor reduction upon combined anti-PD1 and AFL, local, hence broader, application of anti-PD1 holds potential for future treatment of non-aggressive keratinocyte carcinomas.

**Abstract:**

The use of immune checkpoint inhibitors (ICI) is expanding with the approval for advanced/metastatic keratinocyte carcinoma; however, most tumors are non-aggressive. Local administration could broaden ICI, but adequate immune response might require an immune-attractive adjuvant such as ablative fractional laser (AFL). Accordingly, this study aimed to explore intratumoral injection of anti-PD1 with and without AFL in basal cell carcinoma (BCC), exploring anti-PD1 concentration, immune cell infiltration, tumor response, and safety. This open-label, proof-of-concept trial investigated intratumoral anti-PD1 + AFL combination therapy versus anti-PD1 or AFL monotherapy in 28 BCC patients. The primary endpoints were immune cell infiltration evaluated immunohistochemically and clinical tumor response after 3 months. The secondary outcomes were tumoral drug concentration and safety. The most robust response was obtained following intervention with combined anti-PD1+AFL, leading to a ~2.5-fold increase in CD3+ cells (*p* = 0.027), and tumor reduction ≥25% in 73%, including two tumors with complete remission. Upon anti-PD1 monotherapy, a slight decrease in CD3+ cells was observed while a non-significant increase following AFL was seen. Tumor reduction ≥25% was seen in 45% and 50%, respectively, after anti-PD1 and AFL monotherapy. The CD8/CD3 ratio remained unchanged after anti-PD1+AFL and anti-PD1 monotherapy, while AFL led to a decreased ratio. A non-significant decline in the Foxp3/CD3 ratio was observed for all groups. Side-effects were mild with no systemic drug concentration detected. Intratumoral anti-PD1 injection is feasible, and a single exposure to locally injected anti-PD1 with adjuvant AFL increased immune cell infiltration and reduction in BCC with limited side-effects.

## 1. Introduction

Basal cell carcinoma (BCC) is the most common malignancy worldwide and exceeds the prevalence of all other cancers combined [1]. Surgery is first-line treatment while existing local pharmacological therapy is restricted to superficial BCC [2].

Systemic immunotherapy, including immune checkpoint inhibitors (ICI) such as anti-programmed death 1 (anti-PD1), has led to an unprecedented improvement in the treatment of various aggressive cancers, including malignant melanoma (MM) [3]. Still, many patients do not benefit from these treatments due to the failure to activate an effective antitumor immune response [4]. The tumor microenvironment (TME) is crucial in that connection involving a broad variety of suppressive immune cells, tumor-infiltrating lymphocytes (TILs), cancer cells, and fibroblasts, etc. [5].

With FDA approval of cemiplimab for locally advanced and metastatic squamous cell carcinoma (SCC) in 2018 [6] and locally advanced and metastatic BCC in 2021 [7], ICI now includes treatment of keratinocyte carcinomas (KC). Most BCCs, however, are localized skin tumors with low metastasizing potential, arising mostly in elderly patients, many with comorbidities. This, in combination with associated treatment toxicity, restricts ICI to a small subgroup of fit patients with aggressive disease. If locally administered, immunotherapy might be relevant for a larger patient group. As for MM, knowledge of ICI and TME interaction for KC is essential to predict treatment outcome.

Ablative fractional laser (AFL) is widely used in dermatology, and focus on AFL as a mediator of immunological response has gained impact. Recently this has been exploited in pre-clinical trials where adjuvant AFL combined with systemic anti-PD1 improved survival, tumor clearance, and the growth rate in murine BCC [8]. Furthermore, in UV-induced murine SCC, combination therapy with AFL and imiquimod increased the immunological response compared with imiquimod monotherapy [9].

With the overall aim to expand the use of ICI to non-aggressive BCC, this explorative study investigates the local application of anti-PD1. We hypothesize that intratumoral injection of anti-PD1 leads to the increase in immune cells in the TME of BCC impacting tumor growth and that adjuvant AFL further improves immunologic and clinical response. This is explored by the evaluation of tumoral anti-PD1 concentration, immune cell infiltration, tumor reduction, and safety following intratumoral anti-PD1 injection with-and without adjuvant AFL and with AFL monotherapy serving as control.

## 2. Materials and Methods

### 2.1. Study Setup

The study was approved by the local ethics committee (H-19060276), the Danish Medicines Agencies (Eudra-CT 2019-003310-14), and the data protection agency and registered at ClinicalTrials.Gov (NCT04570683). The Good Clinical Practice unit, Copenhagen University, performed external monitoring, and all patients signed an informed consent prior to inclusion.

The study was an open-label, explorative trial with patient recruitment from the Department of Dermatology at Copenhagen University Hospital, Bispebjerg, and from a private dermatology clinic in Hørsholm, Copenhagen, Denmark. The recruitment, intervention, and follow-up period lasted from January 2020 to throughout October 2021. Inclusion criteria were age above 18 and at least one histologically verified BCC with a minimum diameter of 7 mm. All BCC subtypes were eligible for inclusion. Exclusion criteria were recurrent BCC, concomitant treatment with 5-FU or imiquimod, concomitant chemotherapeutic treatment, immunosuppression including use of systemic immunotherapeutic drugs, pregnant or lactating women, allergies to anti-PD1, tendency to form keloids and other skin diseases, or tattoos in the treatment area.

Participants were consecutively recruited for the three study intervention groups. We planned inclusion of 10 patients or BCCs in one group before consecutively starting inclusion for the next intervention group. The three intervention groups consisted of (1) intratumoral anti-PD1 injection with adjuvant AFL, (2) intratumoral anti-PD1 monotherapy, and (3) AFL monotherapy.

Each patient had four study visits. The baseline visit (week 0) included tumor measurement, 4-mm punch biopsy for diagnostic and immunohistochemical assessment, and clinical photography (Canon EOS 60D, Ota City, Japan), with clinical images taken to serve as control and documentation of tumor size and response as well as documentation of local skin reactions. At visit two (week 3), the scheduled intervention was performed. At visit three (week 4), the second 4-mm punch biopsy for immunohistochemistry (IHC) was performed, and a blood sample for detection of systemic nivolumab was taken for a subgroup of participants. If the patient presented with more than one BCC, the second BCC was included for drug detection (without histological or clinical evaluation) of anti-PD1 where the biopsy was taken 1 or 24 h after intervention. At the last visit (week 15), final clinical evaluation, including tumor measurement, clinical photography, and surgical excision or curettage of remaining tumor, was performed (Table 1).

The primary endpoints were (i) local immune cell infiltration and (ii) clinical tumor response, while secondary endpoints were (iii) intratumoral anti-PD1 concentration and (iv) safety of intratumoral anti-PD1 injection, including clinical safety evaluation and systemic detection of anti-PD1.

### 2.2. Interventions

Intratumoral injection with nivolumab 40 mg/4 mL (Bristol-Myers Squibb) was administered at a concentration of 0.1 mL pr cm^2^ treatment area (tumor + 5 mm peritumoral skin). Clinical tumor demarcation with demarcation of a 5 mm safety margin of normal skin was performed by the same experienced dermatologist throughout the study.

If patients were treated with adjuvant AFL, AFL was applied first and immediately followed by intratumoral injection of nivolumab.

AFL was given as a single exposure of 10,600 nm CO2-laser Ultrapulse fractional with a DeepFx handpiece (Lumenis, Inc., Santa Clara, CA, USA) at 100 mJ/microbeam and 5% density. The treatment area included the BCC and a 5 mm rim of peritumoral tissue.

Prior to all interventions, the treatment areas (tumor and 5 mm peritumoral rim) were numbed by local injection of mepivacaine hydrochloride 20 mg/mL + adrenaline 5 µg/mL (Carbocain^®^, AspenNordic, Ballerup, Denmark).

### 2.3. Tumoral Drug Quantification

Quantification of intratumoral nivolumab concentration was performed in a subgroup of patients presenting with more than one BCC and treated with anti-PD1 (with or without AFL), where the extra BCC was included for quantification of nivolumab at 1 or 24 h following intervention. Eight patients participated with a 4 mm punch biopsy for anti-PD1 detection; four BCCs were analyzed following anti-PD1+AFL, (*n* = 2, 1 h, *n* = 2, 24 h), and four BCCs were analyzed after anti-PD1 monotherapy, (*n* = 1, 1 h, *n* = 3, 24 h). Biopsies and BCCs used for drug quantification were not included for analysis of immunological or clinical response.

Quantification of nivolumab was conducted by use of enzyme-linked immunosorbent assay (ELISA). A 4 mm tumor punch biopsy was taken and snap frozen (−80 °C) until analyses. The cutaneous nivolumab concentrations were measured in horizontal sections at skin depths of 0–500, 500–1000, and 1000–1500 μm. The skin sections were cut into thinner section (30 µm) and homogenized in 500 µL PBS using a Retsch MM400 mixer mill (Retsch, Haan, Germany) at 30 Hz for 15 min, followed by 2 h extraction in an up-side-down rotator. Finally, the samples were centrifuged (12 min, 12,000 rcf), supernatant collected, and stored at −80 °C until ELISA analysis. Nivolumab concentrations were quantified using BioSim™Nivolumab (Opdivo^®^) ELISA Kit catalog #E4382 (BioVision, Milpitas, CA, USA). The samples were prepared according to the manufacturer’s protocol with detection performed by use of an 800TS Absorbance microplate reader (BioTek Instruments, Winooski, VT, USA) at 450 nm. To obtain the exact values, the concentrations were determined from the kit-based standard curve and adjusted to volume of the cryosection (0.0063 cm^3^), and the amount of sample was analyzed.

### 2.4. Evaluation of Immunological Response

#### Histological Tissue Processing, Immunohistochemistry, and Digital Photo Analysis

The 4 mm punch biopsies from the skin were formalin-fixed and paraffin-embedded according to standard procedure, and 3-mm sections were cut from the tissue blocks. Each biopsy sample had serial sections placed on microscope slides for staining with hematoxylin and eosin and immunohistochemical staining with antibodies directed towards CD3 (Clone F7.2.38. Agilent/Dako (Santa Clara, CA, USA) (M7254)), CD8 (Clone C8/144B RTU. Agilent/Dako (Santa Clara, CA, USA) (GA623)), Foxp3 (Clone 236A/E7. Invitrogen (Waltham, MA, USA), eBioscience (Waltham, MA, USA) (14-4777-82)), PD1 (Clone NAT105. Cell Marque (Rocklin, CA, USA) (315M-15)), and PD-L1 (Clone 28-8. PharmDX for Autostainer Link 48. Agilent/Dako (Santa Clara, CA, USA) (SK005)) according to the manufacturer’s recommendations. The initial diagnosis of BCC was performed by conventional microscopy by the same pathologist (D.K.). Subsequently, all slides were scanned using a Hamamatsu NanoZoomer XR scanner (Hamamatsu Photonics K.K, Hamamatsu City, Japan) at 40× magnification. All images were analyzed by the same pathologist (J.S.E.) using QuPath-0.3.0. software. A region of interest was defined for each slide, including tumor and peritumoral tissue. The area of basal cell carcinoma cells was subtracted from the total area of the region of interest to obtain the area of the peritumoral connective tissue. In this area, the number of cells with positive staining for each of the immunohistochemical markers CD3, CD8, and FoxP3 was estimated using the algorithm “positive cell detection” of QuPath-0.3.0 and calculated as the number of positive cells per mm^2^ of peritumoral tissue. Staining for PD1 and PD-L1 were assessed by the same pathologist (J.S.E.), and the specimens were scored as either negative (<1% positive tumor cells) or positive (>1% positive tumor cells). The number of TILs staining positive for PD1 was assessed semi-quantitatively for each biopsy and scored to the nearest 10th percentile.

### 2.5. Tumor Response

Tumor response was assessed clinically and histologically. Partial remission was defined as clinical tumor reduction of a minimum of 25% at the three-month follow-up compared with baseline. Complete remission (CR) was defined as clinical as well as histologic analysis without a remaining tumor. Clinical tumor demarcation was performed by the same experienced dermatologist throughout the study. The margin tracings were transferred to a transparent template, allowing for re-identification at follow-up visits.

Change in tumor area was measured in mm^2^ and the size change calculated as reduction in percent. The median of the tumor change in tumor size was calculated for each group and compared.

### 2.6. Safety

Evaluation of local skin reactions (LSR) was performed at day seven after intervention and at final visit, three months following intervention. Six parameters consisting of erythema, edema, crusting, flaking, pustules, and pigmentation were graded on a standardized 5-point severity scale from 0–4, corresponding to none, mild, moderate, prominent, and severe. A total score reflecting overall LSR severity was then calculated based on the sum of all parameters (max sum 24).

The sign of systemic absorption of nivolumab was analyzed by quantification of nivolumab by use of ELISA. In the nivolumab monotherapy group, seven patients contributed a blood sample for detection and quantification of anti-PD1 concentration in blood one week after intervention. The blood samples were centrifuged, the supernatant collected, and samples further prepared and quantified according to manufacturer’s protocol in BioSim™Nivolumab (Opdivo^®^), ELISA Kit catalog #E4382 (BioVision, Milpitas, CA, USA). Detection was performed using an 800TS Absorbance microplate reader (BioTek Instruments, Winooski, VT, USA) at 450 nm. Safety assessment was evaluated as the number of patients experiencing adverse events (AEs) during the study or who had local pain or infection.

### 2.7. Statistical Analyses

The study was designed as an exploratory study; hence, no formal sample size was calculated.

Statistical analyses were performed by use of SPSS version 25 (released 2017, IBM SPSS Statistics for Windows, version 25.0., IBM Corp., Armonk, NY, USA). A Fischer’s exact test was used to compare number of BCCs reaching a 25% tumor reduction or more. The anti-PD1 with adjuvant AFL group was compared one at a time with each of the monotherapy intervention groups. The Wilcoxon signed rank test was used to compare the immune cells before and after intervention in each intervention group, and the Kruskal–Wallis test with Dunn’s multiple comparison test were used to compare change in immune cell ratios between the different intervention groups.

## 3. Results

### 3.1. Baseline Characteristics

We included 28 patients with a total of 39 BCCs consecutively in the three cohorts of the study. Thus, there were 10 patients with 14 BCCs included for anti-PD1 with adjuvant exposure to AFL, while 9 patients with 15 BCCs received anti-PD1 monotherapy, and 9 patients with 10 BCCs had AFL only. All patients completed the study.

Basic demographics of the included patients are shown in Table 2.

### 3.2. Local Tumor Immune Cell Infiltration

Combination therapy with intratumoral anti-PD1 and AFL led to a median 2.5-fold increase in CD3+ cells (*p* = 0.027), and no clear pattern was seen with anti-PD1 without AFL, while AFL monotherapy resulted in a nonsignificant 1.6-fold increase in CD3+ cells (*p* = 0.19) (Figure 1).

When comparing the ratio of CD3+ cells before and after intervention in the three groups, no significant differences were observed between any of the three groups.

While anti-PD1 monotherapy and anti-PD1 + AFL did not change the CD8/CD3 ratio, AFL monotherapy, on the other hand led, to a decrease in the CD8/CD3 ratio (*p* = 0.035).

In terms of Foxp3+ cells, they were quantitatively comparable for all three intervention groups prior to intervention, with a non-significant decline of the FoxP3/CD3 ratio observed for all intervention groups after intervention.

### 3.3. PD1/PD-L1

PD1 expression was seen on TILs in all tumor biopsies from all three intervention groups both in the pre-and post-treatment biopsies. The level of TILs with PD1 expression in the pre-treatment biopsy was comparable for all three intervention groups. Overall, more PD1+ TILs were seen in the pre-treatment tumor biopsies compared with the post-treatment biopsies. This reduction in PD1+ TILs reached statistical significance for tumors treated with anti-PD1 monotherapy (*p* = 0.014) and anti-PD1 with adjuvant AFL (*p* = 0.016) (Figure 2).

Overall, the staining intensity of the PD-L1 was very weak and not quantified.

### 3.4. Tumor Response

Intervention with anti-PD1 and adjuvant AFL led to a tumor reduction of 25% or more in 8/11 (73%) BCCs. In the anti-PD1 monotherapy group, 5/11 (45%) tumors were reduced with 25% or more while this was seen in 5/10 (50%) BCCs in the AFL monotherapy group.

CR, clinical as well as histological, was observed in 2/11 (18%) tumors treated with anti-PD1 and adjuvant AFL and in 2/11 (18%) in the anti-PD1 monotherapy group, while no tumors obtained CR in the AFL monotherapy group. In common for all four tumors with CR was the very high level of peritumoral inflammation postintervention (Figure 3). For the two tumors obtaining CR following anti-PD1 with adjuvant AFL, baseline inflammation was low but increased more than 5-fold postintervention (Figure 1). For the two tumors where CR was observed following anti-PD1 monotherapy, a high inflammatory level was seen prior to as well as after intervention.

### 3.5. Tumoral Drug Concentration

Intratumoral nivolumab was detectable at both time points (1 and 24 h) and following both anti-PD1 with adjuvant AFL and anti-PD1 monotherapy. The concentration of nivolumab 1 h after injection was overall higher than after 24 h for both interventions. There was a tendency for overall higher concentrations following anti-PD1 with adjuvant AFL, particularly pronounced after 24 h. After anti-PD1 with adjuvant AFL, anti-PD1 concentrations were around 20–30 µg/cm^3^ in all three skin depths, compared with 1.8 µg/cm^3^ in the deepest skin layer following anti-PD1 monotherapy (Figure 4)

### 3.6. Safety

#### 3.6.1. Local Skin Reactions

Most patients presented with mild LSR. The LSR consisted mainly of erythema, crusting, and flaking one-week postintervention, with a median composite score of 3/24 (IQR 3–4) in patients treated with anti-PD1 and adjuvant AFL, 2/24 (IQR 0–3) in patients treated with anti-PD1 monotherapy, and 3.5/24 (IQR 2.75–4.75) in the AFL-monotherapy group. At the three-month postintervention, a decline in LSRs was observed and consisted primarily of erythema, crusting, and hyper or hypopigmentation, with a median composite score of, respectively, 1/24 (IQR 0.5–3) (anti-PD1+AFL), 1/24 (IQR 0–2.25) (anti-PD1 monotherapy), and 1.5/24 (IQR 0.75–3) (AFL monotherapy).

#### 3.6.2. Systemic Adverse Events

Since all patients were locally anesthetized prior to AFL exposure and nivolumab injection, no pain was reported in conjunction with intervention. All patients having AFL experienced some oozing immediately following laser treatment lasting up to a maximum of 24 h. One patient undergoing AFL monotherapy experienced local skin infection, with the need for systemic antibiotic treatment. There were no systemic symptoms or other severe AEs during the follow-up, and nivolumab was undetectable in the seven patients that had a blood sample analyzed for drug detection.

## 4. Discussion

With this explorative study on local delivery of ICI in BCC, intratumoral administration of anti-PD1 was shown to be feasible. Upon a single intervention of intratumoral anti-PD1 with adjuvant AFL, CD3 T-cell infiltration in BCC increased. In total, four tumors treated with anti-PD1 with or without adjuvant AFL obtained CR. High peritumoral inflammation was seen in all four, increasing markedly upon adjuvant AFL.

Drug quantification of nivolumab was achievable up to at least 24 h following intratumoral injection. There was a marked decline in tumoral drug concentration from 1 to 24 h. This is consistent with previous in vivo studies [10]. Pharmacokinetic analysis of nivolumab in serum from patients treated every second week with 3 mg/kg reveals plasma concentration ranging from 50 to 100 µg/mL [11]. In our study, the tumoral concentrations of nivolumab 1 h after intervention were around 75 µg/cm^3^, declining to 10–30 µg/cm^3^ after 24 h. Since this is the first clinical trial administering anti-PD1 intratumorally, and since tumoral drug concentrations have not been quantified with systemic administration, comparison with previous studies on tumoral drug concentration is not possible. Further studies on pharmacokinetics and intratumoral administration of anti-PD1 are needed.

While anti-PD1 therapy has been used as treatment of MM since 2014, there is still a lack of knowledge on the interaction with the TME. Given the restricted use of ICI for the treatment of BCC, awareness of biomarkers for this cancer type is scant. BCC grows slowly, and if not ulcerated or infected, clinically they mostly present as non-inflamed, in line with studies on the TME and BCCs indicating an overall attenuated state of immunity [12,13]. In common for the four BCCs where CR was obtained in our study, a high degree of post-treatment peritumoral inflammation was seen. Although speculative, this could signify that the high density of immune cells in the peritumoral tissue is of significance in terms of tumor shrinkage. Casuistically, IHC analysis of two BCCs evolving in a patient while treated with anti-PD1 for lung cancer revealed restricted inflammation with limited CD3+ cells and a lack of human leukocyte antigen class I subunit. The authors propose that the development of BCC during treatment with anti-PD1, hence lack of response, could be due to BCCs in general harboring a “cold” TME [14]. A phase 2 study on cemiplimab for locally advanced BCC has shown response rates of 31%, with only 6% obtaining CR [7]. Accordingly, 69% of treated tumors did not respond to cemiplimab. Given that BCC has one of the highest tumor mutational burdens of all cancers, generally considered as predictive for treatment response to anti-PD1 [15], other causes for the lack of response should be explored. The sparse overall immune cell infiltration observed in relation to BCCs, hence a “cold” TME, could be a factor adding to the lack of response. There is ongoing search for therapeutic approaches to turning “cold” tumors into “hot” tumors, hence improving immunological treatment [16]. Radiotherapy is one of many therapies with emerging application with emphasis on radiation-evoked tumor response [17]. In the present study, the local immune cell infiltration was significantly increased after a single exposure to intratumoral anti-PD1 with adjuvant AFL, while no increase in CD3+ cells was observed with anti-PD1 alone. This could propose the need for adjuvants such as AFL to stimulate an adequate immune reaction upon local injection of anti-PD1. While many physical interventions could serve as a stimulator of the immune systems, AFL was chosen in the current study for several reasons: AFL is well-established in dermatology with the advantage of producing standardized interventions, inducing a grid of microscopic zones of abalted tissue surrounded by the thermally coagulated tissue in the treated skin [18]. Partly due to this thermal injury, a substantial antitumor immune response is stimulated with the recruitment of neutrophils and cytotoxic T-cells [19], induction of immunogenic cell death, and activation of innate and adaptive immune cells [20,21,22]. Further, the concept of laser immunotherapy is gaining impact [23], with promising results in preclinical studies where AFL applied as an adjuvant to systemic anti-PD1 was superior to anti-PD1 alone in terms of overall survival, tumor growth, and immune cell attraction in murine BCC models [8]. Additionally, a substantial increase in tumor clearance and lymphocyte infiltration was seen upon use of AFL as an adjuvant to imiquimod in UV-induced SCC in mice [9]. In oncology, combination therapy represents a conceptual cornerstone to reduce drug resistance and boost therapeutic effectiveness. Exploiting the easily accessible localization of skin tumors by combining physical treatment modalities such as laser as well as other local physical tissue-destructive modalities—e.g., electroporation, cryotherapy, radiotherapy—with ICI might serve as a future combination therapy in dermatologic oncology.

In pre-clinical trials, AFL-induced tumor-specific CD8+ T-cells have been shown [19]. In human BCC, tumor specific T-cell clones have not been found [24], and based on our results, AFL does not induce a CD8+ T-cell response in BCC, at least not in response to a single exposure. Further, in the present study, no correlation between CD8 infiltration and tumor response was observed. Apart from CD8+ T-cells, a decline in PD1+ T-regs is appreciated as a predictor of treatment outcome [25]; however, data are conflicting [26]. We found an overall tendency of a decline in the T-reg/CD3 ratio and PD1+ TILs after intervention, with the latter being statistically significant following intervention with anti-PD1 with or without AFL, but not AFL as monotherapy. Since double staining for PD1 and Foxp3 was not performed, a conclusion on the cell type of the PD1+ TILs cannot be made. Future investigation with double staining of TILs will elucidate whether these lymphocytes are PD1+ T-regs. Competing binding of intratumorally injected anti-PD1 and immunohistochemical staining with an anti-PD1 antibody should be considered as a possible bias when evaluating the level of PD1+ TILs postintervention.

Given the limited patient and tumor number, clinical efficacy is difficult to conclude on, yet the presented data point to increased tumor reduction upon intervention with anti-PD1 and adjuvant AFL, with 73% of tumors obtaining a partial response including two CRs. Essential in considering the clinical outcome is the number of interventions. We explored anti-PD1 and AFL to a single time point only, while in clinical cancer settings, systemic immunotherapy is given at repeated intervals, typically every 2–3 weeks depending on tumor type, patient performance status, side effects, and tumor response [27]. Thus, the repeated intratumoral injection of anti-PD1 might give rise to the continuous local presence of anti-PD1, causing a sustained clinical outcome. The timing of the combination therapy should be further explored in future studies, with AFL and intratumoral anti-PD1 given at staggered time points.

## 5. Conclusions

In summary, with this study we have firstly shown the feasibility of the intratumoral administration of ICI to human BCC with a detectable concentration of anti-PD1 up to at least 24 h. Upon anti-PD1 with adjuvant AFL, an increased immune cell infiltration and high tumor response rate were seen. Given the explorative design, many questions remain unanswered. This study, however, provides new insight to the intratumoral use of ICI in BCC in a clinical setting. This could pave the way for the future broadening of ICI to non-aggressive, easily accessible skin tumors such as KC, where AFL and other physical tissue-destructive modalities could be exploited as immune adjuvants.

## Figures and Tables

**Figure 1 cancers-14-05815-f001:**
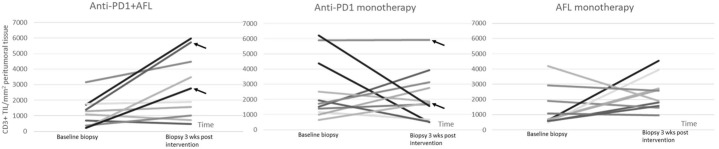
CD3+ T-cells in the three intervention groups before and after intervention. Each line represents one BCC with the slope indicating increase or decrease in CD3+ T-cells before and after intervention, clearly illustrating the overall increase in CD3+ cells following anti-PD1 and adjuvant AFL. The lines marked with arrows illustrate tumors with complete remission. For statistical analysis of CD3+ cells before and after, a Wilcoxon signed rank test was used.

**Figure 2 cancers-14-05815-f002:**
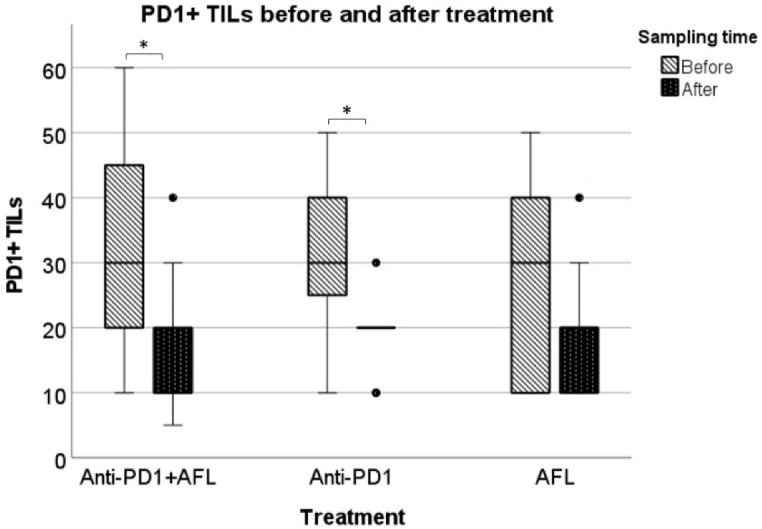
PD1+ TILS in the three intervention groups before and after intervention measured in nearest 10th percentiles. The boxplot illustrates a statistically significant reduction of PD1+ TILs succeeding anti-PD1 with adjuvant AFL and anti-PD1 monotherapy (marked with *) (*n* = 10–11 each intervention group, analyzed by Wilcoxon signed rank test, * *p < 0.05*); ● = outliers.

**Figure 3 cancers-14-05815-f003:**
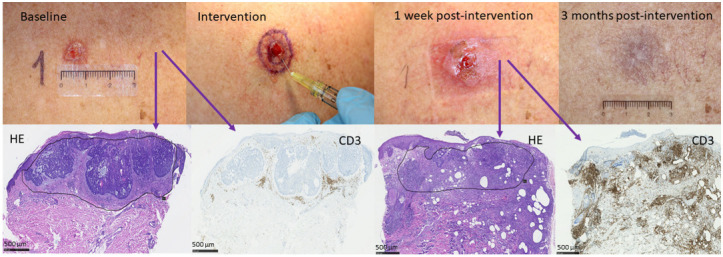
Tumor response to anti-PD1 and adjuvant AFL. Clinical and histological photos of one of the tumors treated with anti-PD1 with adjuvant AFL obtaining CR. The histological photos stained with hematoxylin and eosin (HE) and CD3 at baseline and one week following intervention clearly show a marked increase in peritumoral inflammation, with very subtle inflammation prior to treatment and a dense inflammatory infiltrate one week after combination therapy. Scale bar: 500 µm.

**Figure 4 cancers-14-05815-f004:**
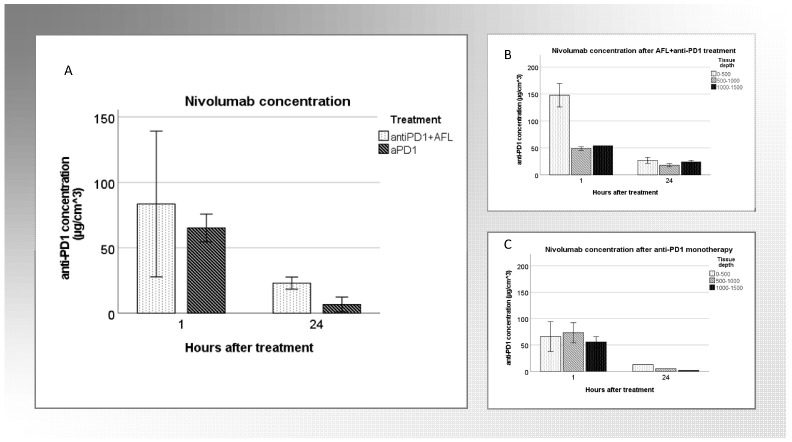
Tumoral anti-PD1 concentration. (**A**) illustrates the drug concentration in the entire skin biopsy (all three skin depths combined) after anti-PD1 + AFL (*n* = 2, 1 h, *n* = 2, 24 h) and anti-PD1 monotherapy (*n* = 1, 1 h, *n* = 3, 24 h) 1 h and 24 h following intervention. (**B**,**C**) show drug concentration in three different skin depths of the biopsy following anti-PD1 and adjuvant AFL (**B**) and anti-PD1 monotherapy (**C**).

**Table 1 cancers-14-05815-t001:** Overview of the study visits.

Baseline, Week 0	Visit 2, Week 3	Visit 3, Week 4	Visit 4, Week 15
Inclusion Tumor measurement Clinical photo tumor Biopsy for histological diagnosis and IHC	Tumor measurement Intervention: ○ AFL+anti-PD1 ○ Anti-PD1 ○ AFL Subgroup: extra tumor biopsy for drug detection (after 1 or 24 h)	Clinical photo Evaluation AEs Biopsy for IHC Subgroup: blood sample for anti-PD1 concentration	Evaluation clinocal response Evaluation AEs Tumor measurement Clinical photo tumor Final excision or curretage

**Table 2 cancers-14-05815-t002:** Basic demographics.

Patients/Tumors	Age, Mean	Sex	BCC Subtype	Anatomical Localization
Overall, Patients, *n* = 28 BCC, *n* = 39	72	F: 10, M: 18	Nodular, *n* =15 Superficial, *n* = 13 Mixed, *n* = 11	Trunk, *n* = 28 Extremities, *n* = 9 Neck, *n* = 2
Anti-PD1+AFL, Patients, *n* = 10 BCC, *n* = 14	73	F: 3, M: 7	Nodular, *n* = 7 Superficial, *n* = 3 Mixed, *n* = 4	Trunk, *n* = 9 Extremities, *n* = 4 Neck, *n* = 1
Anti-PD1, Patients, *n* = 9 BCC, *n* = 15	72	F: 2, M: 7	Nodular, *n* = 2 Superficial, *n* = 8 Mixed, *n* = 5	Trunk, *n* = 11 Extremities, *n* = 3 Neck, *n* = 1
AFL, Patients, *n* = 9 BCC, *n* = 10	70	F: 5, M: 4	Nodular, *n* = 6 Superficial, *n* = 2 Mixed, *n* = 2	Trunk, *n* = 8 Extremities, *n* = 2 Neck, *n* = 0

## Data Availability

No new data were created or analyzed in this study. Data sharing is not applicable to this article.

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
