# Peer review of "Feasibility of Intratumoral Anti-PD1 as Treatment of Human Basal Cell Carcinoma: An Explorative Study with Adjuvant Ablative Fractional Laser"

_cancers, 2022, doi:10.3390/cancers14235815_

Round 1

Reviewer 1 Report

The study ‘A single exposure of intratumoral anti-PD1 with adjuvant ablative fractional laser increases immune cell infiltration in human basal cell carcinoma: a proof-of-concept study’ by S.H. Omland and colleagues reports interesting data on the feasibility of the intralesional administration of ICI in combination with an (adjuvant) AFL session to treat locally confined primary BCC. They conclude that the applied combination could burst antineoplastic immunity, broader the use of anti-PD1 and make the conceptual basis for the future development of alternative therapeutic modalities for non-aggressive keratinocyte carcinomas.

The exploratory study is, overall, adequately designed; however, the rational to use the particular combination should be explained in more detail and the overoptimistic presentation of the findings should be avoided. A potential revision should, at least, address following queries:

-The title should be revised to indicate the ‘explorative’ nature of the study and highlight the main outcome, i.e. ‘feasibility of intratumoral ICI’. Since AFL alone induces local inflammation with the involvement of quite similar cellular composition, explain what is meant with “proof-of-concept”.

-Explain in more details the image analysis procedures for the evaluation of the clinical pictures of the tumors.   

-A main drawback of the study is the interpretation of the statistical inference. Despite being an explorative study, the statistical methodology should be thoroughly revised: The reported p-values should be adapted for multiple tests and the results adequately revised. Note, that all ‘statistical significances’ may wave after correcting for multiple tests. Avoid ‘inferences’ like “…nearly statistically significant difference…” (lines: 223-5).

-Considering the rational of the combination, the authors should explain why they choose AFL within the proposed combination, and not some other physical intervention, taking into consideration that AFL severely denaturates macromolecules, more than other modalities. Discuss, in addition, the timing of the combination: why concomitant application of the two combined modalities?

-Avoid speculative overinterpretations, like “…immune infiltration is …separated from the intratumoral environment by palisading basaloid tumor cells” (lines 320-2): does a functional role of palisading meant?

-Finally, the innovations of the study should be highlighted: Concentrate on reporting feasibility of intratumoral ICI in the clinical setting, options of combination with physical tissue-destructive local interventions, intratumoral ICI pharmacokinetics, role of the accessibility of the skin tumors for the development of non-surgical treatment options for primary neoplasms

Reviewer 2 Report

The manuscript titled “A single exposure of intratumoral anti-PD1 with adjuvant ablative fractional laser increases immune cell infiltration in human basal cell carcinoma: a proof-of-concept study” describes the treatment of basal cell carcinoma via intratumoral administrate immune checkpoint inhibitor, nivolumab. It’s interesting. The followings are some concerns and comments have been pointed out that the authors may want to consider.

1) Line 34: Please use italic p as it refers to a p-value throughout the manuscript.

2) Line 79 study setup: I’d suggest the authors generate a scheme to show patient visits and related treatment clearly.

3) Lines 195-196: Please brief ELISA procedure and include detailed reagents’ information throughout the manuscript.

4) Line 227 Figure 1: a) Please provide high-resolution images. b) Please include at least a brief statistical statement in the figure legend. c) The legend for X-axis is missing.

5) Line 247 Figure 2: a) Please provide a high-resolution image. b) Please include how many samples were involved and the statistical method statement.

6) Line 263 Figure 3: Upper panel most right image, it should be “3 months” instead of “3 mnth”.

7) Line 277 Figure 4: Please state how many samples were involved in the test.

Round 2

Reviewer 1 Report

Congratulations for the interesting study. We look forward reading your follow up reports on non-surgical treatments for primary keratinocyte skin cancers.

Author Response

Thank you very much. We appreciate your time and constructive comments concerning the article.

Reviewer 2 Report

Thank you for the update. Please consider the following comments before publication. Good luck.

1: Line 150: Please include ELISA Kit cat#. Please check throughout the manuscript.

2: Line 262: There is an extra space followed the sign “<”. Or please be consistent with or without a space before and after the sign throughout the manuscript.

3: Please format section 4 and section 5 to meet /cancers/ style.

Author Response

Thank you for your final comments on the manuscript and again thank you for your time reviewing and commenting constructively on the manuscript. 

1. On line 153 and 204 the kit catalog#E4382 has been added 

2. We have deleted the extra space following the sign < on line 269 so it corresponds to the format in the rest of the manuscript

3. We have formatted section 4 and 5